# PROMPT-BASED 3D MOLECULAR DIFFUSION MODELS FOR STRUCTURE-BASED DRUG DESIGN

## ABSTRACT

Generating ligand molecules that bind to specific protein targets via generative models holds substantial promise for advancing structure-based drug design. Existing methods generate molecules from scratch without reference or template ligands, which poses challenges in model optimization and may yield suboptimal outcomes. To address this problem, we propose an innovative prompt-based 3D molecular diffusion model named PROMPTDIFF to facilitate target-aware molecule generation. PROMPTDIFF leverages a curated set of ligand prompts, *i.e.*, those with desired properties such as high binding affinity and synthesizability, to steer the diffusion model towards synthesizing ligands that satisfy design criteria. Specifically, we design a geometric protein-molecule interaction network (*PMINet*), and pretrain it with binding affinity signals to: (i) retrieve target-aware ligand molecules with high binding affinity to serve as prompts, and (ii) incorporate essential protein-ligand binding structures for steering molecular diffusion generation with two effective prompting mechanisms, *i.e.*, *exemplar prompting* and *self prompting*. Empirical studies on CrossDocked2020 dataset show PROMPTDIFF can generate molecules with more realistic 3D structures and achieve state-of-the-art binding affinities towards the protein targets, while maintaining proper molecular properties.

## 1 INTRODUCTION

Designing ligand molecules that can bind to specific protein targets and modulate their function, also known as *structure-based drug design* (SBDD) (Anderson, 2003; Batool et al., 2019), is a fundamental problem in drug discovery and can lead to significant therapeutic benefits. SBDD requires models to synthesize drug-like molecules with stable 3D structures and high binding affinities to the target. Nevertheless, it is challenging and involves massive computational efforts because of the enormous space of synthetically feasible chemicals (Ragoza et al., 2022a) and freedom degree of both compound and protein structures (Hawkins, 2017). Recently, several new generative methods have been proposed for the SBDD task (Li et al., 2021; Luo et al., 2021; Peng et al., 2022; Powers et al., 2022; Ragoza et al., 2022b; Zhang et al., 2023), which learn to generate ligand molecules by modeling the complex spatial and chemical interaction features of the binding site. For instance, some methods adopt autoregressive models (ARMs) (Luo & Ji, 2021; Liu et al., 2022; Peng et al., 2022) and show promising results in SBDD tasks, which generate 3D molecules by iteratively adding atoms or bonds based on the target binding site. However, ARMs tend to suffer from error accumulation, and it is difficult to find an optimal generation order, which are both nontrivial for 3D molecular graphs.

Aiming to address these limitations of ARMs, recent works (Guan et al., 2023a; Schneuing et al., 2022; Lin et al., 2022) adopt diffusion models (Ho et al., 2020) to model the distribution of atom types and positions from a standard Gaussian prior with a post-processing to assign bonds. These diffusion-based SBDD methods develop SE(3)-equivariant diffusion models (Hoogeboom et al., 2022) to capture both local and global spatial interactions between atoms and have achieved better performance than previous autoregressive models. Despite the state-of-the-art performance, it is still difficult for existing diffusion-based methods to generate molecules that satisfies realistic biological metrics such as binding affinity and synthesizability. Moreover, generating from scratch makes the generation process hard to optimize and may lead to suboptimal performance.

To overcome these challenges, we propose a novel **Prompt**-based **Diff**usion model (**PROMPTDIFF**) for SBDD task as in Figure 1. PROMPTDIFF is inspired by the recent significant advancement in machine learning particularly in natural language generation, called prompting or in-context learning (Liu et al., 2023; Gu et al., 2022; Rubin et al., 2022), which enables (large) language models (Brown et al., 2020; OpenAI, 2023) to generalize well to previously-unseen tasks with proper task-specific prompts. Herein, different from previous methods that only rely on the generalization capacity of generative models for new target proteins, PROMPTDIFF explicitly utilizes a small set of target-aware ligand prompts, *i.e.*, molecules with desirable properties such as high binding affinity and synthesizability, to steer the diffusion model toward generating ligands that satisfy design criteria. Specifically, we design a geometric protein-molecule interaction network named ***PMINet*** and pretrain it with binding affinity signals, which is parameterized with SE(3)-equivariant and attention layers to capture both spatial and chemical interaction information between protein-molecule pairs. Then we utilize the pre-trained PMINet to (1) retrieve protein-aware ligand molecules with high binding affinity to serve as molecular prompts, and (2) incorporate essential protein-ligand binding structures for steering molecular diffusion generation with two effective prompting mechanisms, *i.e.*, ***exemplar prompting*** and ***self prompting***, conditioned on both the ligand prompt set and target protein. Extensive experiments on CrossDocked2020 dataset demonstrate the effectiveness of our PROMPTDIFF, achieving new state-of-the-art performance regarding binding-related metrics.

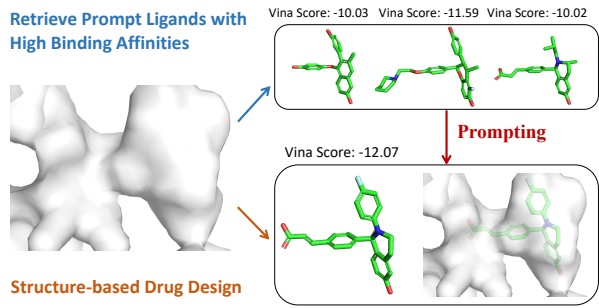

Figure 1: Prompting SBDD with the ligands with high binding affinities towards target protein.

We highlight our main contributions as follows: **(i):** To the best of our knowledge, we for the first time propose a prompt-based 3D molecular diffusion model named PROMPTDIFF for SBDD tasks, steering 3D molecular generation with informative external target-aware prompts. **(ii):** We design two novel prompting mechanisms, *i.e., exemplar prompting* and *self prompting*, to incorporate essential protein-ligand binding structures for facilitating target-aware molecular diffusion generation. **(iii):** Our PROMPTDIFF can generate ligands that not only bind tightly to target pockets but also maintain proper molecular properties. Empirical results on the CrossDocked2020 dataset show that our method achieves **-5.86** Avg. Vina Score and **0.53** Avg. QED score, demonstrating a prominent trade-off between binding-related and property-related metrics.

## 2 RELATED WORK

**Structure-Based Drug Design** As the increasing availability of 3D-structure protein-ligand data (Kinnings et al., 2011), structure-based drug design (SBDD) becomes a hot research area and it aims to generate diverse molecules with high binding affinity to specific protein targets (Luo et al., 2021; Yang et al., 2022; Schneuing et al., 2022; Tan et al., 2022). Early attempts learn to generate SMILES strings or 2D molecular graphs given protein contexts (Skalic et al., 2019; Xu et al., 2021a). However, it is uncertain whether the resulting compounds with generated strings or graphs could really fit the geometric landscape of the 3D structural pockets. More works start to involve 3D structures of both proteins and molecules (Li et al., 2021; Ragoza et al., 2022b; Zhang et al., 2023). Luo et al. (2021), Liu et al. (2022), and Peng et al. (2022) adopt autoregressive models to generate 3D molecules in an atom-wise manner. Recently, powerful diffusion models (Sohl-Dickstein et al., 2015; Song & Ermon, 2019; Ho et al., 2020) begin to play a role in SBDD, and have achieved promising generation results with non-autoregressive sampling (Lin et al., 2022; Schneuing et al., 2022; Guan et al., 2023a). TargetDiff (Guan et al., 2023a), DiffBP (Lin et al., 2022), and DiffSBDD (Schneuing et al., 2022) utilize E(n)-equivariant GNNs (Satorras et al., 2021) to parameterize conditional diffusion models for protein-aware 3D molecular generation. Despite progress, existing methods mainly generate molecules from scratch without informative template or reference ligands for unseen target proteins, which may lead to hard optimization and poor binding affinity. In this paper, PROMPTDIFF for the first time utilize the external ligands with high binding affinity to steer molecular diffusion generation.

**Prompt-Based Controllable Generation** The concept of prompt has been well studied for exploiting the generalization ability of generative models, including natural language processing (Liu et al., 2023; Gu et al., 2022; Rubin et al., 2022), and computer vision (Tsimpoukelli et al., 2021; Alayrac et al., 2022; Gan et al., 2022; Jia et al., 2022). Numerous works have explored techniques to adapt models to novel tasks using a few examples as prompts. Some methods (Chen et al., 2022; Rubin et al., 2022) take input as prompt and retrieve similar examples for prompts augmentation. For example, EPR (Rubin et al., 2022) utilizes a dense retriever to retrieve training examples as prompts for sequence-to-sequence generation. Other methods (Alayrac et al., 2022; Brown et al., 2020; Ouyang et al., 2022) directly construct informative task demonstrations as prompts and perform in-context learning (Min et al., 2022). For example, Flamingo (Alayrac et al., 2022) organizes interleaved text and images as task prompt, and effectively advance few/zero-shot visual question answering. Recently, in the field of drug discovery, RetMol (Wang et al., 2022) employ a retrieval-based framework to control 2D molecule generation for ligand-based drug design (LBDD) (Bacilieri & Moro, 2006). In contrast, our pioneering contribution is to discover proper protein-aware 3D molecule prompts for the purpose of addressing SBDD tasks, which is the first to incorporate complex cross-modal (protein-molecule) geometric interactions in prompt design, thereby introducing a new dimension to the field.

## 3 Preliminary

**Notations** The SBDD task from the perspective of generative models can be defined as generating ligand molecules which can bind to a given protein binding site. The target (protein) and ligand molecule can be represented as $\mathcal{P} = \{(\boldsymbol{x}_P^{(i)}, \boldsymbol{v}_P^{(i)})\}_{i=1}^{N_P}$ and $\mathcal{M} = \{(\boldsymbol{x}_M^{(i)}, \boldsymbol{v}_M^{(i)})\}_{i=1}^{N_M}$, respectively. Here $N_P$ (resp. $N_M$) refers to the number of atoms of the protein $\mathcal{P}$ (resp. the ligand molecule $\mathcal{M}$). $\boldsymbol{x} \in \mathbb{R}^3$ and $\boldsymbol{v} \in \mathbb{R}^K$ denote the position and type of the atom respectively. In the sequel, matrices are denoted by uppercase boldface. For a matrix $\mathbf{X}$, $\mathbf{x}_i$ denotes the vector on its $i$-th row, and $\mathbf{X}_{1:N}$ denotes the submatrix comprising its 1-st to $N$-th rows. For brevity, the ligand molecule is denoted as $\mathbf{M} = [\mathbf{X}_M, \mathbf{V}_M]$ where $\mathbf{X}_M \in \mathbb{R}^{N_M \times 3}$ and $\mathbf{V}_M \in \mathbb{R}^{N_M \times K}$, and the protein is denoted as $\mathbf{P} = [\mathbf{X}_P, \mathbf{V}_P]$ where $\mathbf{X}_P \in \mathbb{R}^{N_P \times 3}$ and $\mathbf{V}_P \in \mathbb{R}^{N_P \times K}$. The task can be formulated as modeling the conditional distribution $p(\mathbf{M}|\mathbf{P})$.

**DDPMs in SBDD** Denoising Diffusion Probabilistic Models (DDPMs) (Ho et al., 2020) equipped with SE(3)-invariant prior and SE(3)-equivariant transition kernel have been applied on the SBDD task (Guan et al., 2023a; Schneuing et al., 2022; Lin et al., 2022). More specifically, the types and positions of the ligand molecule are modeled by DDPM, while the number of atoms $N_M$ is usually sampled from an empirical distribution (Hoogeboom et al., 2022; Guan et al., 2023a) or predicted by a neural network (Lin et al., 2022), and the bonds are determined as post-processing. More concrete details about diffusion models in SBDD can be found in Appendix B.

## 4 Methods

We propose PROMPTDIFF, a novel prompt-based diffusion framework (as demonstrated in Figure 2) for target-aware 3D molecule generation. We first design PMINet with equivariant and attention layers to capture complex 3D geometric interaction information between proteins and ligands, and pretrain it by binding affinity signals (Sec. 4.1). Then, we discover protein-aware ligand prompts with the pre-trained PMINet in the molecular database (Sec. 4.2). Finally, we propose self prompting and exemplar prompting to utilize the ligand prompts with the pre-trained PMINet for facilitating protein-aware 3D molecular generation (Sec. 4.3).

### 4.1 Modeling Protein-Ligand Interactions with PMINet

To capture the interactions between proteins and ligand molecules, we introduce a 3D protein-ligand interaction network (namely PMINet) to model the binding affinity of protein-ligand pairs, consisting of SE(3)-equivariant neural networks (Satorras et al., 2021) and cross-attention layers (Borgeaud et al., 2022; Hou et al., 2019). Two shallow SE(3)-equivariant neural networks are applied on the graphs of the protein $\mathcal{G}_P$ and ligand molecule $\mathcal{G}_M$, respectively, to extract both chemical and geometric

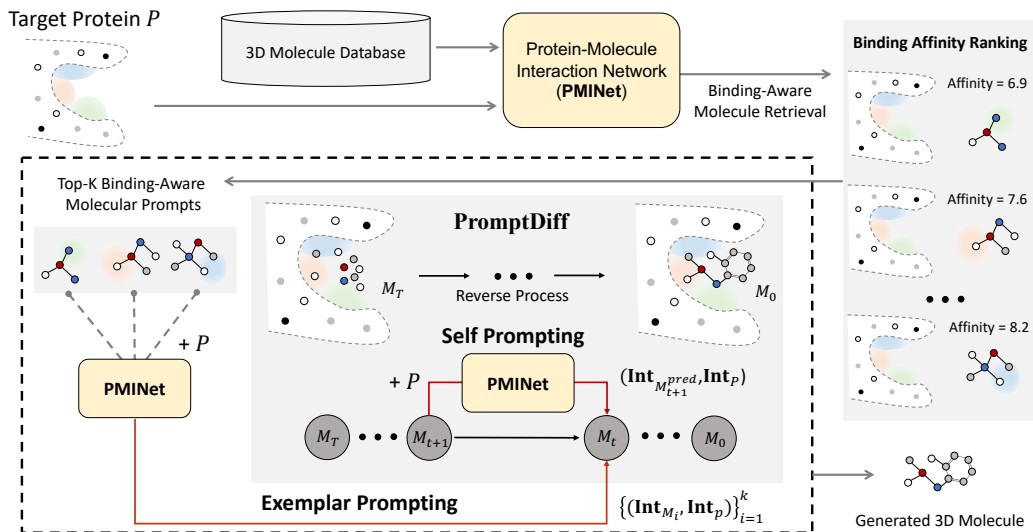

Figure 2: The overall schematic diagram of PROMPTDIFF. It first utilizes the pre-trained **PMINet** (in Sec. 4.1) to retrieve binding-aware ligand prompts (in Sec. 4.2), and then steers the target-aware 3D molecular diffusion generation with **self and exemplar prompting** (in Sec. 4.3). $(\mathbf{Int}_{M_{t+1}^{\mathrm{pred}}}, \mathbf{Int}_P)$ and $\{(\mathbf{Int}_{M_i}, \mathbf{Int}_P)\}_{i=1}^k$ are binding-specific molecular and protein embeddings enhanced by PMINet.

information. Given a ligand graph $\mathcal{G}_M$, the $l$-th SE(3)-equivariant layer works as follows:

$$\mathbf{h}_{M,i}^{l+1} = \mathbf{h}_{M,i}^l + \sum_{j \in \mathcal{N}_M(i)} f_{M,h}^l \left( \left\| \mathbf{x}_{M,i}^l - \mathbf{x}_{M,j}^l \right\|, \mathbf{h}_{M,i}^l, \mathbf{h}_{M,j}^l \right), \tag{1}$$

$$\mathbf{x}_{M,i}^{l+1} = \mathbf{x}_{M,i}^l + \sum_{j \in \mathcal{N}_M(i)} \left( \mathbf{x}_{M,i}^l - \mathbf{x}_{M,j}^l \right) f_{M,x}^l \left( \left\| \mathbf{x}_{M,i}^l - \mathbf{x}_{M,j}^l \right\|, \mathbf{h}_{M,i}^{l+1}, \mathbf{h}_{M,j}^{l+1} \right), \tag{2}$$

where $\mathbf{h}_{M,i}^{l+1} \in \mathbb{R}^d$ and $\mathbf{x}_{M,i}^{l+1} \in \mathbb{R}^3$ are the SE(3)-invariant and SE(3)-equivariant hidden states of the atom $i$ of the ligand after the $l$-th SE(3)-equivariant layer, respectively. $\mathcal{N}_M(i)$ stands for the set of neighbors of atom $i$ on $\mathcal{G}_M$, and the initial hidden state $\mathbf{h}_{M,i}^0$ is obtained by an embedding layer that encodes atom information. Given a protein graph $\mathcal{G}_P$, $\mathbf{h}_{P,i}^l$ and $\mathbf{x}_{P,i}^l$ can be derived in the same way.

In PMINet, an atom-wise cross-attention based interaction layer is proposed to learn the inter-molecule interactions between protein-ligand pairs, which essentially accounts for the binding affinity. Due to the unknown binding poses in practice, only the SE(3)-invariant features $\mathbf{H}_M^L \in \mathbb{R}^{N_M \times d}$ and $\mathbf{H}_P^L \in \mathbb{R}^{N_P \times d}$ (whose $i$-th rows are $\mathbf{h}_{M,i}^L$ and $\mathbf{h}_{P,i}^L$ respectively) are used as inputs to the cross-attention (Vaswani et al., 2017) layer for extracting binding-aware interactive representations:

$$\mathbf{Int}_M = \mathrm{softmax}((\mathbf{W}_Q \mathbf{H}_M^L)(\mathbf{W}_K \mathbf{H}_P^L)^T) \mathbf{W}_V \mathbf{H}_P^L, \tag{3}$$

$$\mathbf{Int}_P = \mathrm{softmax}((\mathbf{W}_Q \mathbf{H}_P^L)(\mathbf{W}_K \mathbf{H}_M^L)^T) \mathbf{W}_V \mathbf{H}_M^L, \tag{4}$$

where $\mathbf{W}_Q, \mathbf{W}_K, \mathbf{W}_V$ are learnable projection matrices. The enhanced representations of the protein and molecule (i.e., $\mathbf{Int}_P$ and $\mathbf{Int}_M$) are further aggregated into a global feature to predict their binding affinity: $S_{\mathrm{Aff}}(\mathcal{M}, \mathcal{P}) := \mathrm{PMINet}(\mathcal{M}, \mathcal{P})$. Please refer to Appendix D.1 for pre-training details.

## 4.2 CONSTRUCTING TARGET-AWARE LIGAND PROMPTS

Inspired by the recent success of prompt-based controllable generation (Liu et al., 2023), we hope to design an effective prompting strategy for structure-based drug design to guide the generation of target-aware ligands with desired properties. Therefore, we leverage the structure-based protein-ligand interaction prior learned by PMINet to select the top candidates (i.e., ligands with high-binding affinity) most suitable for prompting the later target-aware molecule design.

More concretely, given a target $\mathcal{P}$ and an external database of ligands $\mathcal{D} := \{\mathcal{M}_i\}_{i=1}^N$, we use the pre-trained $\mathrm{PMINet}(\cdot, \mathcal{P})$ introduced in Sec. 4.1 to scan the database and retrieve the ligands with

top-$k$ high predicted binding affinity to this target as prompts:

$$\mathcal{D}(\mathcal{P}, k) := \text{top}_k(\{\mathcal{M}_i\}_{i=1}^N, \text{PMINet}(\cdot, \mathcal{P})). \tag{5}$$

Here we denote the prompt pool as $\mathcal{D}(\mathcal{P}, k)$. The ligand molecules in the external database are real, so they are expected to provide some valid substructures as references and thus promote the validity of the generated ligand molecules. For example, the ligands in the prompt pool can be viewed as probes to explore how the target interacts with ligands, which is supposed to offer useful clues for molecule generation. Due to their high binding affinity, they can potentially reveal the critical locations (*e.g.*, promising hydrogen donors or acceptors) to support strong inter-molecular forces.

At time step $t$, we extend the prompt pool to $\{\mathcal{M}_{t+1}^{\text{pred}}\} \cup \mathcal{D}(\mathcal{P}, k)$, where $\mathcal{M}_{t+1}^{\text{pred}}$ denotes the predicted atom positions and types (*i.e.*, estimated $[\hat{\mathbf{X}}_0, \hat{\mathbf{V}}_0]$) at time step $t + 1$. This can be regarded as self-prompting, different from self-conditioning (Chen et al., 2023), which will be described next.

### 4.3 PROMPT-BASED 3D EQUIVARIANT MOLECULAR DIFFUSION

In this subsection, we describe how to introduce the bingding-aware ligand prompts into the design of the neural network $\phi_\theta$ which predicts (*i.e.*, reconstructs) $[\mathbf{X}_0, \mathbf{V}_0]$ in the reverse generation process (we highlight the critical parts of our prompting mechanisms in violet):

$$[\hat{\mathbf{X}}_0, \hat{\mathbf{V}}_0] = \phi_\theta([\mathbf{X}_t, \mathbf{V}_t], t, \mathbf{P}, \{\mathcal{M}_{t+1}^{\text{pred}}\} \cup \mathcal{D}(\mathcal{P}, k)). \tag{6}$$

**Self Prompting** We first extract atom-wise embeddings $\mathbf{H}_M \in \mathbb{R}^{N_M \times d}$ and $\mathbf{H}_P \in \mathbb{R}^{N_P \times d}$ of the ligand molecule being generated and target protein, respectively. The molecule being generated, $\mathcal{M}_{t+1}^{\text{pred}}$, itself is supposed to be a candidate ligand with high binding affinity to the target, especially when $t$ is large (*i.e.*, the generative process nearly ends). To maximize the exploitation of protein-ligand interaction prior emerged in the reverse diffusion trajectories, we leverage the enhanced molecular atom embedding $\mathbf{Int}_{M_{t+1}^{\text{pred}}}$ and protein atom embedding $\mathbf{Int}_P$ produced by the interaction layer of $\text{PMINet}(\mathcal{M}_{t+1}^{\text{pred}}, \mathcal{P})$ to self-prompt the generative process as follows:

$$\text{Self Prompting}: \mathbf{H}_M' = \text{MLP}\left([\mathbf{H}_M, \mathbf{Int}_{M_{t+1}^{\text{pred}}}]\right), \; \mathbf{H}_P' = \text{MLP}\left([\mathbf{H}_P, \mathbf{Int}_P]\right). \tag{7}$$

In training, due to the inaccessibility of $\mathcal{M}_{t+1}^{\text{pred}}$, we directly use ground truth molecule $\mathcal{M}$ to substitute it in a teacher-forcing fashion. We illustrate more insights about self prompting in Appendix C.

**Exemplar Prompting** We further propose exemplar prompting to leverage the exemplar ligands for steering the reverse generation process. The pre-trained PMINet is reused here to extract interactive structural context information between the target protein $\mathcal{P}$ and the ligands in prompt pool $\mathcal{D}(\mathcal{P}, k)$ to enhance the target protein representations:

$$\text{Exemplar Prompting}: \mathbf{H}_P'' = \text{Pool}(\{\text{MLP}([\mathbf{H}_P', \mathbf{Int}_P^i])\}_{i=1}^k), \tag{8}$$

where $\mathbf{Int}_P^i$ is the binding-aware protein feature produced by the interaction layer of $\text{PMINet}(\mathcal{M}_i, \mathcal{P})$ (as Equation (4)) and $\mathcal{M}_i$ is the $i$-th exemplar ligand in the prompt pool $\mathcal{D}(\mathcal{M}, k)$.

Besides, in order to prompt the molecular diffusion generation with possible binding structures in exemplar ligands, we merge the enhanced embeddings of exemplar ligands and generated molecules with a trainable cross attention mechanism:

$$\text{Exemplar Prompting}: \mathbf{H}_M'' = \text{Pool}(\{\text{softmax}((\mathbf{W}_Q \mathbf{H}_M')(\mathbf{W}_K \mathbf{Int}_{M_i})^T)\mathbf{W}_V \mathbf{Int}_{M_i}\}_{i=1}^k), \tag{9}$$

where $\mathbf{Int}_{M_i}$ is the binding-aware exemplar ligand feature produced by the interaction layer of $\text{PMINet}(\mathcal{M}_i, \mathcal{P})$ (as Equation (3)). Our PROMPTDIFF uses self prompting and exemplar prompting to sufficiently leverage both the informative binding prior in external ligands and the protein-aware interaction context for facilitating 3D equivariant molecular diffusion generation.

**3D Equivariant Molecular Diffusion** We then apply an SE(3)-equivariant neural network on the $k$-nn graph of the protein-ligand complex (denoted as $\mathbf{C} = [\![\mathbf{M}, \mathbf{P}]\!]$, where $[\![\cdot]\!]$ denotes concatenation

along the first dimension) to learn the atom-wise protein-molecule interactions in generative process. The SE(3)-invariant hidden state $\mathbf{H}_C$ and SE(3)-equivariant positions $\mathbf{X}_C$ are updated as follows:

$$\mathbf{h}_{C,i}^{l+1} = \mathbf{h}_{C,i}^l + \sum_{j \in \mathcal{N}_C(i)} f_{C,h}^l \left( \left\| \mathbf{x}_{C,i}^l - \mathbf{x}_{C,j}^l \right\|, \mathbf{h}_{C,i}^l, \mathbf{h}_{C,j}^l, \mathbf{e}_{C,ij} \right) \tag{10}$$

$$\mathbf{x}_{C,i}^{l+1} = \mathbf{x}_{C,i}^l + \sum_{j \in \mathcal{N}_C(i)} \left( \mathbf{x}_{C,i}^l - \mathbf{x}_{C,j}^l \right) f_{C,x}^l \left( \left\| \mathbf{x}_{C,i}^l - \mathbf{x}_{C,j}^l \right\|, \mathbf{h}_{C,i}^{l+1}, \mathbf{h}_{C,j}^{l+1}, \mathbf{e}_{C,ij} \right) \cdot \mathbb{1}_{\text{mol}} \tag{11}$$

where $\mathcal{N}_C(i)$ stands for the set of $k$-nearest neighbors of atom $i$ on the protein-ligand complex graph, $\mathbf{e}_{C,ij}$ indicates the atom $i$ and atom $j$ are both protein atoms or both ligand atoms or one protein atom and one ligand atom, and $\mathbb{1}_{\text{mol}}$ is the ligand atom mask since the protein atom coordinates are known and thus supposed to remain unchanged during this update. We let $\mathbf{H}_C^0 := \llbracket \mathbf{H}_M'', \mathbf{H}_P'' \rrbracket$ to incorporate the information contained in the prompt pool $\{\mathcal{M}_{t+1}^*\} \cup \mathcal{D}(\mathcal{P}, k)$. Finally, we use $\hat{\mathbf{V}}_0 = \text{softmax}(\text{MLP}(\mathbf{H}_{C,1:N_M}^L))$ and $\hat{\mathbf{X}}_0 = \mathbf{X}_{C,1:N_M}^L$ as the final prediction. Shifting the Center of Mass (CoM) of protein atoms to zero (Xu et al., 2021b; Hoogeboom et al., 2022; Guan et al., 2023a) and the design of EGNN (Satorras et al., 2021) ensure SE(3)-equivariance of the reverse transition kernel $p_\theta(\mathbf{X}_{t-1} | \mathbf{X}_t, \mathbf{X}_P)$. Prompting the generative process only augments the SE(3)-invariant hidden states without breaking SE(3)-equivariance.

**Training and Sampling** We leave the details about the loss function and summarize the training procedure of PROMPTDIFF in Appendix A. Given a protein $\mathcal{P}$, the ligand molecules can be sampled as Algorithm 1. The differences from previous molecular diffusion models, are highlighted in violet.

---

**Algorithm 1** Sampling Procedure of PROMPTDIFF

---

**Input:** The protein binding site $\mathcal{P}$, the learned model $\phi_\theta$, external databse $\mathcal{D}$, pre-trained PMINet, number of exemplar ligands in each prompt pool $k$
**Output:** Generated ligand molecule $\mathcal{M}$ that binds to the protein pocket $\mathcal{P}$
1: Sample the number of atoms $N_M$ of the ligand molecule $\mathcal{M}$ as described in Sec. 3
2: Move CoM of protein atoms to zero
3: Sample initial ligand atom coordinates $\mathbf{x}_T$ and atom types $\mathbf{v}_T$
4: Let $\mathbf{M}^* := [\mathbf{0}, \mathbf{0}]$
5: Embed $\mathbf{V}_P$ into $\mathbf{H}_P$
6: **for** $t$ in $T, T-1, \ldots, 1$ **do**
7:   Embed $\mathbf{V}_t$ into $\mathbf{H}_M$
8:   Obtain features $\mathbf{H}_M'$ and $\mathbf{H}_P'$ with self-prompting based on $\mathbf{M}^*$ (Equation (7))
9:   Obtain enhanced protein atom feature $\mathbf{H}_P''$ prompted by $\mathcal{D}(\mathcal{P}, k)$ (Equation (8))
10:   Obtain enhanced ligand atom feature $\mathbf{H}_M''$ prompted by $\mathcal{D}(\mathcal{P}, k)$ (Equation (9))
11:   Predict $[\hat{\mathbf{X}}_0, \hat{\mathbf{V}}_0]$ from $[\mathbf{X}_t, \mathbf{H}_M'']$ and $[\mathbf{X}_P, \mathbf{H}_P'']$ (Equations (10) and (11))
12:   Sample $\mathbf{X}_{t-1}$ and $\mathbf{V}_{t-1}$ from the posterior $p_\theta(\mathbf{M}_{t-1} | \mathbf{M}_t, \mathbf{P})$ (Equation (16))
13:   Let $\mathbf{M}^* := [\hat{\mathbf{X}}_0, \hat{\mathbf{V}}_0]$
14: **end for**

---

## 5 EXPERIMENTS

**Datasets and Baseline Methods** To pretrain PMINet with binding affinity signals, we use the PDBbind v2016 dataset (Liu et al., 2015), which is most frequently used in binding-affinity prediction tasks. Specifically, 3767 complexes are selected as training set, and the other 290 complexes are selected as testing set. As for molecular generation, following the previous work (Luo et al., 2021; Peng et al., 2022; Guan et al., 2023a), we train and evaluate PROMPTDIFF on the CrossDocked2020 dataset (Francoeur et al., 2020). We follow the same data preparation and splitting as (Luo et al., 2021), where the 22.5 million docked binding complexes are refined to high-quality docking poses (RMSD between the docked pose and the ground truth $< 1$Å) and diverse proteins (sequence identity $< 30\%$). This produces $100,000$ protein-ligand pairs for training and 100 proteins for testing.

We randomly choose 128 ligands from training set for retrieval, and select the ligand of top-1 predicted binding affinity as prompt for each protein. We compare our model with recent representative methods

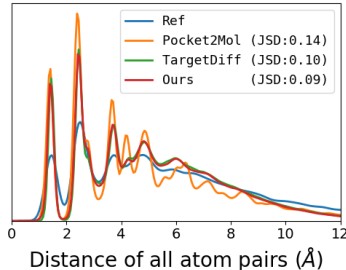

Figure 3: Comparing the distribution for distances of all-atom for reference molecules in the test set (blue) and model generated molecules (color). Jensen-Shannon divergence (JSD) between two distributions is reported.

| Bond | liGAN | AR | Pocket2 Mol | Target Diff | Decomp Diff | ours |
|------|-------|------|-------------|-------------|-------------|------|
| C−C | 0.601 | 0.609 | 0.496 | 0.369 | **0.359** | 0.396 |
| C=C | 0.665 | 0.620 | 0.561 | 0.505 | 0.537 | **0.249** |
| C−N | 0.634 | 0.474 | 0.416 | 0.363 | 0.344 | **0.307** |
| C=N | 0.749 | 0.635 | 0.629 | 0.550 | 0.584 | **0.280** |
| C−O | 0.656 | 0.492 | 0.454 | 0.421 | 0.376 | **0.373** |
| C=O | 0.661 | 0.558 | 0.516 | 0.461 | **0.374** | 0.442 |
| C:C | 0.497 | 0.451 | 0.416 | 0.263 | 0.251 | **0.213** |
| C:N | 0.638 | 0.552 | 0.487 | 0.235 | 0.269 | **0.264** |

Table 1: Jensen-Shannon divergence between bond distance distributions of the reference molecules and the generated molecules, and lower values indicate better performances. "-", "=", and ":" represent single, double, and aromatic bonds, respectively.

for SBDD. **LiGAN** (Ragoza et al., 2022a) is a conditional VAE model trained on an atomic density grid representation of protein-ligand structures. **AR** (Luo et al., 2021) and **Pocket2Mol** (Peng et al., 2022) are autoregressive schemes that generate 3D molecules atoms conditioned on the protein pocket and previous generated atoms. **TargetDiff** (Guan et al., 2023a) and **DecomposeDiff** (Guan et al., 2023b) are recent state-of-the-art non-autoregressive diffusion-based molecular diffusion models.

**Evaluation**   We comprehensively evaluate the generated molecules from three perspectives: **molecular structures**, **target binding affinity** and **molecular properties**. In terms of **molecular structures**, we calculate the Jensen-Shannon divergences (JSD) in empirical distributions of atom/bond distances between the generated molecules and the reference ones. To estimate the **target binding affinity**, following previous work (Luo et al., 2021; Ragoza et al., 2022b; Guan et al., 2023a), we adopt AutoDock Vina (Eberhardt et al., 2021) to compute and report the mean and median of binding-related metrics, including *Vina Score*, *Vina Min*, *Vina Dock* and *High Affinity*. Vina Score directly estimates the binding affinity based on the generated 3D molecules; Vina Min performs a local structure minimization before estimation; Vina Dock involves an additional re-docking process and reflects the best possible binding affinity; High affinity measures the ratio of how many generated molecules binds better than the reference molecule per test protein. To evaluate **molecular properties**, we utilize the *QED*, *SA*, *Diversity* as metrics following (Luo et al., 2021; Ragoza et al., 2022a). QED is a simple quantitative estimation of drug-likeness combining several desirable molecular properties; SA (synthesize accessibility) is a measure estimation of the difficulty of synthesizing the ligands; Diversity is computed as average pairwise dissimilarity between all generated ligands.

## 5.1   MAIN RESULTS

**Generated Molecular Structures**   We compare our PROMPTDIFF and the representative methods in terms of molecular structures. We plot the all-atom pairwise distance distribution of the generated molecules in Figure 3. PROMPTDIFF achieves lowest JSD 0.09 to reference in the all-atom pairwise distance distribution of the generated molecules, which is better than two strong baseline methods Pocket2Mol and TargetDiff, indicating it effectively captures real atomic distances. We compute different bond distributions of the generated molecules and compare them against the corresponding reference empirical distributions in Tab. 1. Our model has a comparable performance with DecompDiff and substantially outperforms other baselines across most major bond types, indicating the great potential of PROMPTDIFF for generating stable molecular structures. We attribute this to our prompt mechanisms which directly provides realistic 3D ligand templates for steering molecule generation.

**Target Binding Affinity and Molecule Properties**   We evaluate the effectiveness of PROMPTDIFF in terms of binding affinity. We can see in Tab. 2 that our PROMPTDIFF outperforms baselines in binding-related metrics. Specifically, PROMPTDIFF surpasses strong autoregressive method Pocket2Mol by a large margin of **14.0%** and **38.5%** in Avg. and Med. Vina Score, and surpasses strong diffusion-based method DecompDiff by **3.4%** and **7.8%** in Avg. and Med. Vina Score. In terms of high-affinity binder, we find that on average **66.8%** of the PROMPTDIFF molecules show better binding affinity than the reference molecule, which is significantly better than other baselines. These gains demonstrate that the proposed PROMPTDIFF effectively utilizes the external protein-ligand interactions to enable generating molecules with higher target binding affinity.

Table 2: Summary of different properties of reference molecules and molecules generated by our model and other non-diffusion and diffusion-based baselines. (↑) / (↓) denotes a larger / smaller number is better. Top 2 results are highlighted with **bold text** and underlined text, respectively.

| Methods | | Vina Score (↓) | | Vina Min (↓) | | Vina Dock (↓) | | High Affinity (↑) | | QED (↑) | | SA (↑) | | Diversity (↑) | |
|---|---|---|---|---|---|---|---|---|---|---|---|---|---|---|---|
| | | Avg. | Med. | Avg. | Med. | Avg. | Med. | Avg. | Med. | Avg. | Med. | Avg. | Med. | Avg. | Med. |
| Reference | | -6.36 | -6.46 | -6.71 | -6.49 | -7.45 | -7.26 | - | - | 0.48 | 0.47 | 0.73 | 0.74 | - | - |
| Compare with Non-Diffusion | LiGAN | - | - | - | - | -6.33 | -6.20 | 21.1% | 11.1% | 0.39 | 0.39 | 0.59 | 0.57 | 0.66 | 0.67 |
| | GraphBP | - | - | - | - | -4.80 | -4.70 | 14.2% | 6.7% | 0.43 | 0.45 | 0.49 | 0.48 | **0.79** | **0.78** |
| | AR | -5.75 | -5.64 | -6.18 | -5.88 | -6.75 | -6.62 | 37.9% | 31.0% | 0.51 | 0.50 | 0.63 | 0.63 | 0.70 | 0.70 |
| | Pocket2Mol | -5.14 | -4.70 | -6.42 | -5.82 | -7.15 | -6.79 | 48.4% | 51.0% | **0.56** | **0.57** | 0.74 | **0.75** | 0.69 | 0.71 |
| | **PROMPTDIFF** | **-5.86** | **-6.51** | **-7.14** | **-7.27** | **-8.33** | **-8.49** | 66.8% | 73.9% | 0.53 | 0.54 | 0.58 | 0.58 | 0.72 | 0.72 |
| Compare with Diffusion | TargetDiff | -5.47 | -6.30 | -6.64 | -6.83 | -7.80 | -7.91 | 58.1% | 59.1% | 0.48 | 0.48 | 0.58 | 0.58 | 0.72 | 0.71 |
| | DecompDiff | -5.67 | -6.04 | -7.04 | -7.09 | **-8.39** | **-8.43** | 64.4% | 71.0% | 0.45 | 0.43 | **0.61** | **0.60** | 0.68 | 0.68 |
| | **PROMPTDIFF** | **-5.86** | **-6.51** | **-7.14** | **-7.27** | -8.33 | -8.49 | 66.8% | 73.9% | **0.53** | **0.54** | 0.58 | 0.58 | **0.74** | **0.72** |

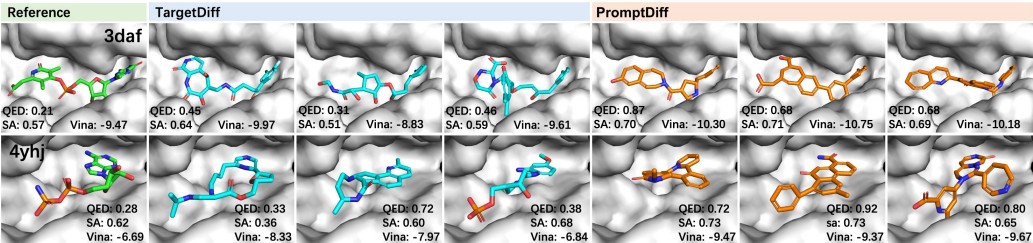

Figure 4: Reference ligands and generated ligand molecules of TargetDiff (Guan et al., 2023a) and PROMPTDIFF for 3daf (top row) and 4yhj (bottom row). We report QED, SA, and Vina Dock score for each molecule.

Ideally, using whole training set for retrieval can significantly improve the both binding- and property-related metrics as demonstrated in Tab. 4, but it would increase computational burden. Thus we randomly choose 128 ligand molecules for retrieval for all experiments. Moreover, we can see a trade-off between property-related metrics (QED and SA) and binding-related metrics in previous methods. TargetDiff and DecompDiff perform better than AR and Pocket2Mol in binding-related metrics, but fall behind them in QED and SA scores. In contrast, our PROMPTDIFF not only achieves the state-of-the-art binding-related scores but also maintains proper QED score, achieving a better trade-off than TargetDiff and DecompDiff. Nevertheless, we put less emphasis on QED and SA because they are often applied as rough screening metrics in real drug discovery scenarios, and it would be fine as long as they are within a reasonable range. Figure 4 shows some examples of generated ligand molecules and their properties. The molecules generated by our model have valid structures and reasonable binding poses to the target, which are supposed to be promising candidate ligands. We visualize more examples of generated molecules in Appendix F.

## 5.2 MODEL ANALYSIS

**Influence of Self Prompting and Exemplar Prompting**    We investigate the impact of self prompting and exemplar prompting mechanisms of PROMPTDIFF. We showcase the efficacy of self prompting and exemplar prompting in our PROMPTDIFF, and put the results in Tab. 3. In particular, we remove our prompting mechanisms from PROMPTDIFF and use it as baseline. We observe that simply applying prompting mechanism without our pre-trained PMINet even hurt the generation performance, because it does not include informative protein-ligand interactions for self refinement. In contrast, our self prompting significantly improve both binding- and property-related metrics due to the informative interaction knowledge brought by our pre-trained PMINet. Besides, our exemplar prompting also has a notable improvement over baseline, revealing that the external target-aware prompts indeed provide a suitable reference and facilitate the optimization for molecular generation. Exemplar prompting does not help much in property-related metrics, because we only focus on useful binding structures in exemplars.

Furthermore, using both self and exemplar prompting achieves the best binding-related metrics, demonstrating the effectiveness of the two complementary prompting mechanisms in PROMPTDIFF. Figures 5 and 6 provide both quantitative and qualitative analyses of the similarity between the generated ligands and their corresponding prompts, indicating that the prompts indeed serve as

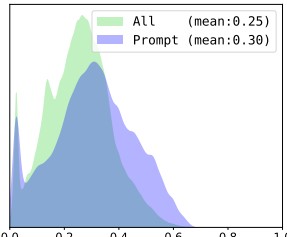

Figure 5: The distributions of Tanimoto similarity between generated ligands and (a) all ligands in database, and (b) corresponding prompts.

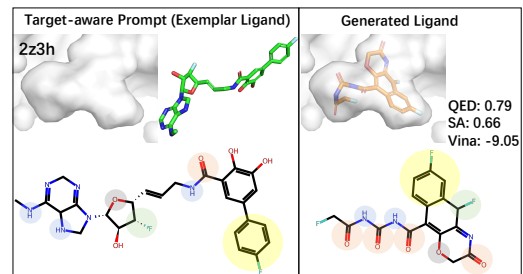

Figure 6: Example of a generated ligand molecule and its corresponding prompt. Important substructures shared by these two molecules are highlighted in the same colors.

Table 3: Ablation studies of self prompting and exemplar prompting in PROMPTDIFF.

| Methods | Vina Score (↓) | | Vina Min (↓) | | Vina Dock (↓) | | High Affinity (↑) | | QED (↑) | | SA (↑) | |
|---|---|---|---|---|---|---|---|---|---|---|---|---|
| | Avg. | Med. | Avg. | Med. | Avg. | Med. | Avg. | Med. | Avg. | Med. | Avg. | Med. |
| baseline | -5.04 | -5.75 | -6.38 | -6.52 | -7.55 | -7.72 | 54.2% | 54.1% | 0.46 | 0.46 | 0.57 | 0.57 |
| prompt without PMINet | -4.14 | -5.64 | -6.11 | -6.36 | -7.60 | -7.67 | 54.9% | 57.5% | 0.47 | 0.48 | 0.57 | 0.57 |
| + self prompting | -4.91 | -6.03 | -6.53 | -6.82 | -7.95 | -8.14 | 61.5% | 64.4% | **0.52** | **0.53** | **0.59** | **0.58** |
| + exemplar prompting | -5.39 | -6.28 | -6.40 | -6.67 | -8.14 | -8.37 | 64.0% | 71.5% | 0.51 | 0.52 | 0.57 | 0.57 |
| + both prompting | **-5.86** | **-6.51** | **-7.14** | **-7.27** | **-8.33** | **-8.49** | **66.8%** | **73.9%** | 0.53 | 0.54 | 0.58 | 0.58 |

exemplars and guide the generation. More specifically, Figure 6 shows the model may automatically select critical substructures for high binding affinity to the target from the exemplar prompt ligands and reassemble them in proper positions to generate molecules.

**Effect of Molecule Retrieval Database** We investigate the influence of the 3D molecule database for retrieval in PROMPTDIFF through ablation study on two variables $n$ and $k$, where $n$ denotes the size of the molecule database and $k$ denotes the number of prompt exemplars. From the results in Tab. 4, we observe that larger $n$ can benefit PROMPTDIFF in terms of binding-related metrics, because higher diversity allows for more binding-related cues (substructures) that can prompt the generation process. Simply increasing $k$ does not have an obvious improvement because leveraging more molecule prompts would also introduce more noises into generation process. Kindly note that the retrieval database is fixed during both training and testing, and we further evaluate the robustness of our model to the choices of retrieval database in Appendix E.

Table 4: The effect of the size $n$ of molecule database. (↑) / (↓) denotes a larger / smaller number is better. Top 2 results are highlighted with **bold text** and underlined text, respectively.

| Methods | Vina Score (↓) | | Vina Min (↓) | | Vina Dock (↓) | | High Affinity (↑) | | QED (↑) | | SA (↑) | |
|---|---|---|---|---|---|---|---|---|---|---|---|---|
| | Avg. | Med. | Avg. | Med. | Avg. | Med. | Avg. | Med. | Avg. | Med. | Avg. | Med. |
| $n = 32$ | -5.67 | -6.21 | -7.01 | -7.13 | -8.13 | -8.30 | 65.2% | 72.0% | 0.51 | 0.52 | 0.56 | 0.57 |
| $n = 64$ | -5.74 | -6.39 | -7.07 | -7.19 | -8.21 | -8.35 | 65.9% | 72.8% | 0.52 | 0.53 | 0.57 | **0.58** |
| $n = 128$ | **-5.86** | **-6.51** | **-7.14** | **-7.27** | **-8.33** | **-8.49** | **66.8%** | **73.9%** | **0.53** | **0.54** | **0.58** | **0.58** |
| $k = 1$ | -5.86 | -6.51 | -7.14 | -7.27 | -8.33 | **-8.49** | **66.8%** | 73.9% | **0.53** | **0.54** | **0.58** | **0.58** |
| $k = 2$ | -5.88 | **-6.62** | -7.25 | **-7.29** | -8.34 | -8.42 | 66.6% | **74.4%** | **0.53** | **0.54** | 0.57 | 0.56 |
| $k = 3$ | **-5.97** | -6.57 | **-7.31** | -7.22 | **-8.40** | -8.42 | 66.2% | 73.5% | 0.52 | 0.52 | **0.58** | **0.59** |

## 6 CONCLUSION

In this work, we for the first time propose a prompt-based 3D molecular diffusion model PROMPTDIFF for SBDD. We leverage the target-aware prompt ligands to enhance the 3D molecular diffusion generation with effective self prompting and exemplar prompting mechanisms, significantly improving the binding affinity measured by Vina while maintaining proper molecular properties. For future work, we will incorporate other binding-related information into the generation process.

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

# A    TRAINING PROCEDURE

To train PROMPTDIFF (*i.e.*, optimize the evidence lower bound induced by PROMPTDIFF), we use the same objective function as Guan et al. (2023a). The atom position loss and atom type loss at time step $t - 1$ are defined as follows respectively:

$$L_{t-1}^{(x)} = \frac{1}{2\tilde{\beta}_t^2} \sum_{i=1}^{N_M} \|\tilde{\boldsymbol{\mu}}(\mathbf{x}_{i,t}, \mathbf{x}_{i,0}) - \tilde{\boldsymbol{\mu}}(\mathbf{x}_{i,t}, \hat{\mathbf{x}}_{i,0})\|^2 = \gamma_t \sum_{i=1}^{N_M} \|\mathbf{x}_{i,0} - \hat{\mathbf{x}}_{i,0}\|; \quad (12)$$

$$L_{t-1}^{(v)} = \sum_{i=1}^{N_M} \sum_{k=1}^{K} \tilde{\boldsymbol{c}}(\mathbf{v}_{i,t}, \mathbf{v}_{i,0})_k \log \frac{\tilde{\boldsymbol{c}}(\mathbf{v}_{i,t}, \mathbf{v}_{i,0})_k}{\tilde{\boldsymbol{c}}(\mathbf{v}_{i,t}, \hat{\mathbf{v}}_{i,0})_k}; \quad (13)$$

where $\hat{\mathbf{X}}_0$ and $\hat{\mathbf{V}}_0$ are predicted from $\mathbf{X}_t$ and $\mathbf{V}_t$, and $\gamma_t = \frac{\bar{\alpha}_{t-1}\beta_t^2}{2\tilde{\beta}_t^2(1-\bar{\alpha}_t)^2}$. Kindly recall that $\mathbf{x}_{i,t}$, $\mathbf{v}_{i,t}$, $\hat{\mathbf{x}}_{i,0}$, and $\hat{\mathbf{v}}_{i,0}$ correspond to the $i$-th row of $\mathbf{X}_t$, $\mathbf{V}_t$, $\hat{\mathbf{X}}_0$, and $\hat{\mathbf{V}}_0$, respectively.

The final loss combines the above two losses with a hyperparameter $\lambda$ as: $L = L_{t-1}^{(x)} + \lambda L_{t-1}^{(v)}$. We summarize the training procedure of PROMPTDIFF in Algorithm 2 and highlight the differences from its counterpart, TargetDiff (Guan et al., 2023a), in violet.

---

**Algorithm 2** Training Procedure of PROMPTDIFF

---

**Input:** Protein-ligand binding dataset $\{\mathcal{P}, \mathcal{M}\}_{i=1}^N$, neural network $\phi_\theta$, external database $\mathcal{D}$, pre-trained PMINet, number of exemplar ligands in each prompt pool $k$
1:  Screen $\{\mathcal{P}, \mathcal{M}\}_{i=1}^N$ and retrieve ligands with top-$k$ high binding affinity from $\mathcal{D}$ using PMINet to obtain $\{\mathcal{P}, \mathcal{M}, \mathcal{D}(\mathcal{P}, k)\}_{i=1}^N$ as described in Sec. 4.2
2:  **while** $\phi_\theta$ not converge **do**
3:      Sample diffusion time $t \in \mathcal{U}(0, \dots, T)$
4:      Move the complex to make CoM of protein atoms zero
5:      Perturb $[\mathbf{X}_0, \mathbf{V}_0]$ to obtain $[\mathbf{X}_t, \mathbf{V}_t]$
6:      Embed $\mathbf{V}_t$ into $\mathbf{H}_M^0$, and embed $\mathbf{V}_P$ into $\mathbf{H}_P^0$
7:      Obtain features $\mathbf{H}_M'$ and $\mathbf{H}_P'$ with self-prompting based on $[\mathbf{X}_0, \mathbf{V}_0]$        (Equation (7))
8:      Obtain enhanced protein atom feature $\mathbf{H}_P''$ prompted by $\mathcal{D}(\mathcal{P}, k)$        (Equation (8))
9:      Obtain enhanced ligand atom feature $\mathbf{H}_M''$ prompted by $\mathcal{D}(\mathcal{P}, k)$        (Equation (9))
10:     Predict $[\hat{\mathbf{X}}_0, \hat{\mathbf{V}}_0]$ from $[\mathbf{X}_t, \mathbf{H}_M'']$ and $[\mathbf{X}_P, \mathbf{H}_P'']$        (Equations (10) and (11))
11:     Compute loss $L$ with $[\hat{\mathbf{X}}_0, \hat{\mathbf{V}}_0]$ and $[\mathbf{X}_M, \mathbf{V}_M]$        (Equations (12) and (13))
12:     Update $\theta$ by minimizing $L$
13: **end while**

---

# B    TARGET-AWARE MOLECULAR DIFFUSION

In the forward diffusion process, a small Gaussian noise is gradually injected into data as a Markov chain. Because noises are only added on ligand molecules but not proteins in the diffusion process, we denote the atom positions and types of the ligand molecule at time step $t$ as $\mathbf{X}_t$ and $\mathbf{V}_t$ and omit the subscript $M$ without ambiguity. The diffusion transition kernel can be defined as follows:

$$q(\mathbf{M}_t|\mathbf{M}_{t-1}, \mathbf{P}) = \prod_{i=1}^{N_M} \mathcal{N}(\mathbf{x}_{i,t}; \sqrt{1-\beta_t}\mathbf{x}_{i,t-1}, \beta_t \boldsymbol{I}) \cdot \mathcal{C}(\mathbf{v}_{i,t}|(1-\beta_t)\mathbf{v}_{i,t-1} + \beta_t/K), \quad (14)$$

where $\mathcal{N}$ and $\mathcal{C}$ stand for the Gaussian and categorical distribution respectively, $\beta_t$ is defined by fixed variance schedules. The corresponding posterior can be analytically derived as follows:

$$q(\mathbf{M}_{t-1}|\mathbf{M}_t, \mathbf{M}_0, \mathbf{P}) = \prod_{i=1}^{N_M} \mathcal{N}(\mathbf{x}_{i,t-1}; \tilde{\boldsymbol{\mu}}(\mathbf{x}_{i,t}, \mathbf{x}_{i,0}), \tilde{\beta}_t \boldsymbol{I}) \cdot \mathcal{C}(\mathbf{v}_{i,t-1}|\tilde{\boldsymbol{c}}(\mathbf{v}_{i,t}, \mathbf{v}_{i,0})), \quad (15)$$

where $\tilde{\boldsymbol{\mu}}(\mathbf{x}_{i,t}, \mathbf{x}_{i,0}) = \frac{\sqrt{\bar{\alpha}_{t-1}}\beta_t}{1-\bar{\alpha}_t}\mathbf{x}_{i,0} + \frac{\sqrt{\alpha_t}(1-\bar{\alpha}_{t-1})}{1-\bar{\alpha}_t}\mathbf{x}_{i,t}$, $\tilde{\beta}_t = \frac{1-\bar{\alpha}_{t-1}}{1-\bar{\alpha}_t}\beta_t$, $\alpha_t = 1-\beta_t$, $\bar{\alpha}_t = \prod_{s=1}^t \alpha_s$, $\tilde{\boldsymbol{c}}(\mathbf{v}_{i,t}, \mathbf{v}_{i,0}) = \frac{\boldsymbol{c}^*}{\sum_{k=1}^K c_k^*}$, and $\boldsymbol{c}^*(\mathbf{v}_{i,t}, \mathbf{v}_{i,0}) = [\alpha_t \mathbf{v}_{i,t} + (1-\alpha_t)/K] \odot [\bar{\alpha}_{t-1}\mathbf{v}_{i,0} + (1-\bar{\alpha}_{t-1})/K]$.

In the approximated reverse process, also known as the generative process, a neural network parameterized by $\theta$ learns to recover data by iteratively denoising. The reverse transition kernel can be approximated with predicted atom types $\hat{\mathbf{v}}_{i,0}$ and atom positions $\hat{\mathbf{x}}_{i,0}$ as follows:

$$p_\theta(\mathbf{M}_{t-1}|\mathbf{M}_t, \mathbf{P}) = \prod_{i=1}^{N_M} \mathcal{N}(\mathbf{x}_{i,t-1}; \tilde{\boldsymbol{\mu}}(\mathbf{x}_{i,t}, \hat{\mathbf{x}}_{i,0}), \tilde{\beta}_t \boldsymbol{I}) \cdot \mathcal{C}(\mathbf{v}_{i,t-1}|\tilde{\boldsymbol{c}}(\mathbf{v}_{i,t}, \hat{\mathbf{v}}_{i,0})). \tag{16}$$

## C  SELF PROMPTING

Here we offer more insights about self prompting. During sampling, at time step $t$, we utilize the protein-ligand interaction information embedded in $\mathrm{PMINet}(\mathcal{M}_{t+1}^{\mathrm{pred}}, \mathcal{P})$ to guide the generative process itself. For efficient training, given a protein-ligand pair $(\mathcal{P}, \mathcal{M})$, due to inaccessibility of $\mathcal{M}_{t+1}^{\mathrm{pred}}$, we directly replace it with the ground truth ligand molecule $\mathcal{M}$.

Because the training objective is to generate $\mathcal{M}$ given $\mathcal{P}$, a straightforward question is whether using $\mathcal{M}$ as input would provide a shortcut signal for the model and lead to its training collapse. Thanks to the design of PMINet, the model cannot naively rely on the input $\mathcal{M}$ to generate $\mathcal{M}$. More specifically, in PMINet, $\mathcal{M}$ and $\mathcal{P}$ are first input into two separate EGNNs, and only the produced SE(3)-invariant features, which are agnostic to the coordinate systems, are further input in the cross-attention layer to capture the protein-ligand interaction information. Thus, in the output produced by the cross-attention layer of PMINet, the relative positions and poses between the protein and ligand molecule in the physical world are eliminated, and only the protein-ligand interaction information in the feature space is kept. This means no shortcut signal is left for the model during training and the model still needs to normally learn from the protein context to generate the ligand molecule.

## D  IMPLEMENTATION DETAILS

### D.1  DETAILS OF PMINET

**Input Initialization**  To represent each protein atom, we use a one-hot element indicator {H, C, N, O, S, Se} and one-hot amino acid type indicator (20 types). Similarly, we represent each ligand atom with a one-hot element indicator {C, N, O, F, P, S, Cl}. Additionally, we introduce a one-dimensional flag to indicate whether the atoms belong to the protein or ligand. Two 1-layer MLPs are used to map the input protein and ligand into 128-dim latent spaces respectively.

**Model Architectures**  We aim to use PMINet to model the complex 3D interactions between the atoms of proteins and ligands. To achieve this, we use two shallow SE(3)-equivariant neural networks for geometric message passing on the fully-connected graphs of the protein and ligand, respectively. We then apply a cross attention layer to the paired protein-ligand graph for learning the inter-molecule interactions. Finally, we use a sum-pooling layer to extract a global representation of the protein-ligand pair by pooling all atom nodes. And a two-layer MLP is introduced to predict the binding affinity $S_{\mathrm{Aff}}$. More details about the model architecture are provided in Tab. 6.

**Training Details**  During the training, we use the Mean Squared Error (MSE) loss with respect to the difference between the predicted and ground truth binding affinity scores as the optimization objective. We train PMINet on a single NVIDIA V100 GPU, and we use the Adam as our optimizer with learning rate 0.001, $betas = (0.95, 0.999)$, batch size 16. The experiments are conducted on PDBBind v2016 dataset as mentioned in main text.

**Evaluation of PMINet**  We evaluate PMINet's effectiveness in predicting binding affinity, and compare it with a baseline model which is specifically designed for binding affinity prediction, *i.e.*, GraphDTA (Nguyen et al., 2021). Following Li et al. (2021), we select Root Mean Square Error (RMSE), Mean Absolute Error (MAE), Pearson's correlation coefficient (R) and the standard deviation (SD) in regression to measure the prediction error. The testing results are present in Tab. 5, indicating the rationality of our model design. In PROMPTDIFF, we use PMINet to serve as a binding-aware ligand retriever and a protein-ligand interaction information extractor.

Table 5: Performance of PMINet in binding affinity prediction. (↑) / (↓) denotes a larger / smaller number is better. Top 1 results are highlighted with **bold text**.

| Methods | RMSE (↓) | MAE (↓) | SD (↓) | R (↑) |
|---------|----------|---------|--------|-------|
| GraphDTA | 1.562 | **1.191** | 1.558 | 0.697 |
| PMINet | **1.554** | 1.193 | **1.520** | **0.716** |

## D.2   DETAILS OF PROMPT-BASED DIFFUSION MODEL

**Input Initialization**   For balancing the computational burden and the generation performance, we construct the retrieval 3D molecule database by randomly sampling 128 ligand molecules from the training set of CrossDocked2020, and the database is fixed in both training and testing. Then for each target protein, we use PMINet to scan the retrieval database and only select one ligand with the top-1 binding affinity predicted as the prompt molecule. To represent each protein atom, we use a one-hot element indicator {H, C, N, O, S, Se} and one-hot amino acid type indicator (20 types). Similarly, we represent each ligand atom using a one-hot element indicator {C, N, O, F, P, S, Cl}. Additionally, we introduce a one-dimensional flag to indicate whether the atoms belong to the protein or ligand. Two 1-layer MLPs are introduced to map the inputs of protein and ligand into 128-dim spaces respectively. For representing the connection between atoms, we introduce a 4-dim one-hot vector to indicate four bond types: bond between protein atoms, ligand atoms, protein-ligand atoms or ligand-protein atoms. And we introduce distance embeddings by using the distance with radial basis functions located at 20 centers between 0 Å and 10 Å. Finally we calculate the outer products of distance embedding and bond types to obtain the edge features.

**Model Architectures**   At the $l$-th layer, we dynamically construct the protein-ligand complex with a $k$-nearest neighbors (knn) graph based on coordinates of the given protein and the ligand from previous layer. In practice, we set the number of neighbors $k_n = 32$. As mentioned in Sec. 4.3, we apply an SE(3)-equivariant neural network for message passing. The 9-layer equivariant neural network consists of Transformer layers with 128-dim hidden layer and 16 attention heads. Following Guan et al. (2023a), in the diffusion process, we select the fixed sigmoid $\beta$ schedule with $\beta_1 = 1e-7$ and $\beta_T = 2e-3$ as variance schedule for atom coordinates, and the cosine $\beta$ schedule with $s = 0.01$ for atom types. The number of diffusion steps are set to 1000.

**Training Details**   We use the Adam as our optimizer with learning rate $0.001$, $betas = (0.95, 0.999)$, batch size 4 and clipped gradient norm 8. We balance the atom type loss and atom position loss by multiplying a scaling factor $\lambda = 100$ on the atom type loss. We train the parameterized diffusion denoising model of our PROMPTDIFF on a single NVIDIA V100 GPU, and it could converge within 200k steps.

Table 6: Details of both PMINet and Prompt-based Diffusion Model in our PROMPTDIFF

| Network | Module | Backbone | Input Dimensions | Output Dimensions | Blocks |
|---------|--------|----------|------------------|-------------------|--------|
| PMINet | Protein Encoder | EGNN | $N_P \times 128$ | $N_P \times 128$ | 2 |
|  | Ligand Encoder | EGNN | $N_M \times 128$ | $N_M \times 128$ | 2 |
|  | Interaction Layer | Graph Attention Layer | $(N_P + N_M) \times 128$ | $(N_P + N_M) \times 128$ | 1 |
|  | Pooling | Sum-pooling | $(N_P + N_M) \times 128$ | $1 \times 128$ | 1 |
| Prompt-based Diffusion Model | Position Dynamics | Transformer | $(N_P + N_M) \times 3$ | $(N_P + N_M) \times 3$ | 9 |
|  | Atom Type Dynamics | Transformer | $(N_P + N_M) \times 128$ | $(N_P + N_M) \times 128$ | 9 |
|  | Protein Fusion Layer | MLP | $N_P \times (128 + 128)$ | $N_P \times 128$ | 1 |
|  | Ligand Fusion Layer | CrossAttention | $\{N_M \times 128, N_{\text{prompt}} \times 128\}$ | $N_M \times 128$ | 1 |

## E   ABLATION STUDY

**Robustness to The Choice of Retrieval Dataset**   In our experiments, we use the same 3D molecule database for searching target-aware molecular prompts in training and testing. Here we randomly choose two molecule databases for training and testing, and investigate how well our PROMPTDIFF

adapts to novel prompts in testing stage. To simplify the experiments, we set the size of the molecule database to 128 and set the number of ligand prompts to 1. As presented in Tab. 7, even with a new molecule database in testing phase, PROMPTDIFF can still find useful ligand prompts and achieve the same promising generation results. It reveals that our PROMPTDIFF has an excellent generalization ability and is not reliant on a specific molecule database for molecule retrieval.

In addition, we want to investigate whether reselecting a different molecular database would have an impact on the generation results. We randomly choose another two set of molecules and use them to train our PROMPTDIFF. As demonstrated in Tab. 7, reselecting a different molecular database does not have an obvious impact on the generation results, because PROMPTDIFF only focuses on useful substructures in prompts not the whole structure, demonstrating the robustness of our method.

Table 7: Results of evaluating the model robustness. (↑) / (↓) denotes a larger / smaller number is better. Top 2 results are highlighted with **bold text** and underlined text, respectively. "reselect-1" and "reselect-2" denote two reselected molecular databases.

| Methods | Vina Score (↓) | | Vina Min (↓) | | Vina Dock (↓) | | High Affinity (↑) | | QED (↑) | | SA (↑) | | Diversity (↑) | |
|---|---|---|---|---|---|---|---|---|---|---|---|---|---|---|
| | Avg. | Med. | Avg. | Med. | Avg. | Med. | Avg. | Med. | Avg. | Med. | Avg. | Med. | Avg. | Med. |
| PROMPTDIFF | **-5.86** | -6.51 | -7.14 | **-7.27** | -8.33 | **-8.49** | **66.8%** | 73.9% | **0.53** | **0.54** | 0.58 | 0.58 | **0.72** | 0.72 |
| with novel set | -5.83 | **-6.55** | -7.16 | -7.19 | -8.26 | -8.45 | 66.2% | 73.3% | **0.53** | **0.54** | 0.60 | 0.59 | **0.72** | 0.71 |
| reselect-1 | -5.85 | -6.53 | **-7.21** | -7.23 | -8.33 | -8.44 | 66.5% | 73.5% | 0.52 | 0.53 | **0.61** | **0.61** | 0.71 | 0.72 |
| reselect-2 | -5.84 | -6.49 | -7.16 | -7.25 | **-8.34** | -8.47 | 66.6% | **74.1%** | **0.53** | **0.54** | 0.58 | 0.58 | **0.72** | **0.73** |

## F   MORE VISUALIZATION RESULTS

We provide the visualization of more ligand molecules generated by PROMPTDIFF, comparing to both reference and TargetDiff (Guan et al., 2023a), as shown in Figure 7.

We provide the source files containing generated molecules and the evaluation code that can reproduce the results in Tab. 2 in the supplementary material.

We are committed to open source the code of training and inference as well as the pre-trained model upon paper acceptance.

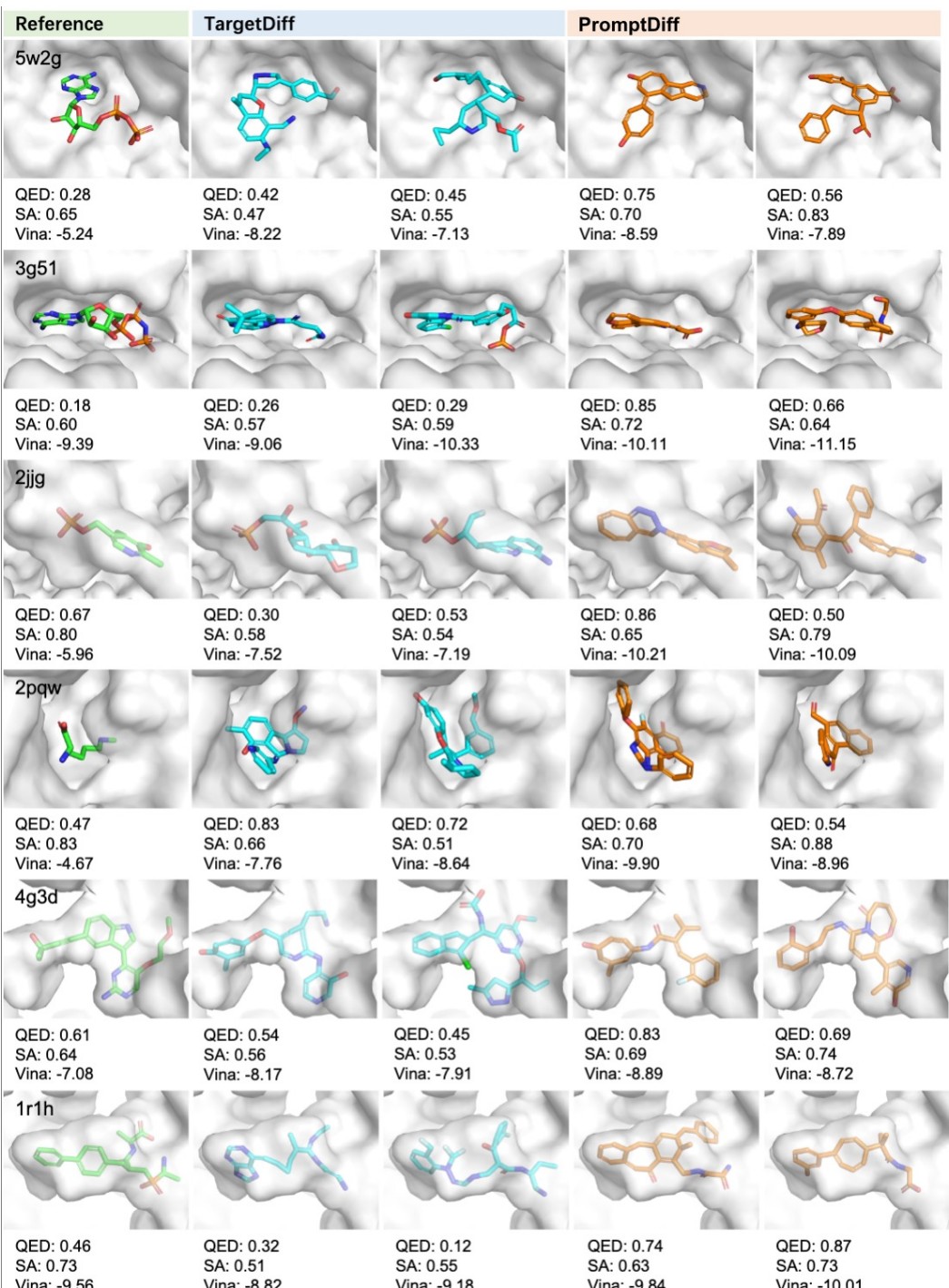

Figure 7: Examples of generated ligands. Carbon atoms in reference ligands, ligands generated by TargetDiff (Guan et al., 2023a) and our model are visualized in green, cyan, and orange respectively. We report QED, SA, and Vina Dock score for each molecule.

