# OpenReview forum: "Prompt-based 3D Molecular Diffusion Models for  Structure-based Drug Design"
_ICLR.cc/2024/Conference — Submitted to ICLR 2024_

### Official Review · Reviewer_gTfj · 2023-10-26

**Soundness:** 2 fair
**Presentation:** 2 fair
**Contribution:** 1 poor
**Rating:** 3
**Confidence:** 4

**Summary:**

The paper proposes a diffusion model for de novo structure-based ligand generation. Compared to the existing methods, current model has two additional mechanisms: self prompting and exemplar prompting that aim to make model generate molecules that interact with the target pocket with higher affinity. To enable exemplar prompting, authors train a separate scoring model to predict binding affinity. The performance seems to reach the state of the art.

**Strengths:**

The idea about additional prompts for the model using exemplary molecules seems promising, as such prompts suggest the model what can potentially work for a specific target. Figure 6 gives a very nice illustration of this idea. Besides, the model with prompting seems to work a bit better in terms of Vina scores, however, I would be interested in looking at the standard deviation of these scores.

**Weaknesses:**

Even though prompting sounds like a nice idea in general, the overall contribution of the current paper seems to me rather limited. Besides, the scope of application is quite unclear. Below I elaborate on these two main concerns and discuss several smaller issues:

1. First of all, for me the scenario in which this model can be applied is not clear. Is it correct that PromptDiff can be used only for pockets that already have known binders? What if I want to generate a molecule for the pocket which has never been targeted before?

2. Why cannot existing scoring methods be used for selecting prompts from the database? What is the reason for creating a new model for predicting binding affinity given that: (a) the proposed model does not seem to suggest any novelty compared to what has been done before, (b) it does not really outperform GraphDTA.

3. In my opinion, the part about exemplar prompting is not clearly explained. Do you consider experimental data? Do you dock molecules to the target? Besides, how exactly do you create a database? Where do you take examples from?

4. Abstract and introduction suggest that you focus on optimizing different molecular properties through prompting while in reality you aim to improve only binding affinity. While you mention synthesizability in the abstract and intro, there is no focus on it in the paper at all. Besides, SA scores of PromptDiff samples are very mediocre. I would suggest to change the wording or to explain better how synthesizability can be addressed by your model and support it by experiments.

5. Algorithm 1, line 1: I did not find the explanation of the sampling procedure of your method in Section 3.

**Questions:**

1. How do train/test sets for PMINet and PromptDiff correspond?

2. Are notations $\mathcal{M}$ and $\boldsymbol{M}=[\boldsymbol{X}, \boldsymbol{V}]$ interchangeable? What's the reason to use different ones?

3. It seems strange for me that adding self-prompting results in degradation of the average Vina score in Table 3. Did you investigate this issue?

4. "_We randomly choose 128 ligands from training set for retrieval, and select the ligand of top-1 predicted binding affinity as prompt for each protein_" – is it 128 random different ligands that you use for all your samples or is it 128 ligands bound to the same pocket? Could you please explain this procedure in more detail?

5. How do you sample initial atom types?

6. What is the difference between self-conditioning [1] and your self-prompting?

**Additional suggestions:**

1. Would be interesting to look at a couple of case studies where one or multiple active compounds are known. How does PromptDiff perform there? Do samples share some parts with these compounds? What are scores of those?

2. Suggested scoring model can be used to score samples so that you get the ranked list of samples.

[1] Chen, T., Zhang, R., & Hinton, G. (2022). Analog bits: Generating discrete data using diffusion models with self-conditioning

---

### Official Review · Reviewer_78qn · 2023-10-30

**Soundness:** 3 good
**Presentation:** 3 good
**Contribution:** 2 fair
**Rating:** 3
**Confidence:** 4

**Summary:**

This paper proposes PROMPTDIFF, a prompt-based 3D molecular diffusion model for generating 3D ligand molecules that bind to specific protein targets. In particular, PROMPTDIFF leverages the target-aware prompt ligands to steer the 3D molecular diffusion models towards generating ligands that satisfy the designed criteria. PROMPTDIFF is compared with recent representative SBDD methods in terms of molecular structures, target binding affinity, and molecular properties.

**Strengths:**

1. This work proposes two novel prompting mechanisms (i.e., self-prompting and exemplar prompting), and the effectiveness has been fully proved through experiments.
2. The experimental results show that PROMPTDIFF can achieve state-of-the-art binding affinities towards the protein targets.

**Weaknesses:**

1.  Inference efficiency: Given a target protein, this method needs to first retrieve target-aware ligand molecules from the database as prompts, and then use the ligand prompts to steer the diffusion model. Therefore, the method will inevitably inherit the efficiency issues of the diffusion model and be even less efficient.
2. More Baselines:
     - Zhang Z, Min Y, Zheng S, et al. Molecule generation for target protein binding with structural motifs, ICLR2023.
     - Zhang Z, Liu Q. Learning Subpocket Prototypes for Generalizable Structure-based Drug Design, ICML2023.
     - Zhang O, Zhang J, Jin J, et al. ResGen is a pocket-aware 3D molecular generation model based on parallel multiscale modeling, Nature Machine Intelligence, 2023: 1-11.
3. Writings: The description of this method is unclear, and the use of symbols is very confusing, especially for the superscript and subscript.
4. Others: The settings in Table 4 are not clear, please add n-k pair information.

**Questions:**

1. Compared with current binding-affinity prediction methods, what’s the advantage of your PMINet? If it can not achieve state-of-the-art performance, why not use better methods to retrieve target-aware ligand molecules?
2. What’s the relationship between ligand prompts and ground truth ligands? Why not directly use the ground truth ligand as prompt?
3. Since the protein target is certain, why enhance its atom feature in PROMPTDIFF?
4. In Table3, how to prompt without PMINet?

---

### Official Review · Reviewer_M3og · 2023-11-02

**Soundness:** 3 good
**Presentation:** 3 good
**Contribution:** 3 good
**Rating:** 6
**Confidence:** 3

**Summary:**

This paper proposes a prompt-based 3D molecular diffusion model, called PomptDiff, that aims to advance structure-based drug design through target-aware generative molecule design.
The main idea is to train a geometric protein-molecule interaction network (PMINet) that is trained based on protein-ligand pairs with known binding affinity, which is then used to select "target-aware" molecules with high predicted binding affinity against the given target and use them as effective prompts to guide the generative diffusion process.
The paper shows that PromptDiff can effectively generate molecules with high binding affinities to protein targets and good molecular properties by incorporating the pre-trained PMINet and the use of effective exemplar prompting and self-prompting techniques.

**Strengths:**

The paper introduces a "target-aware" molecular design approach based on the use of a small set of molecular prompts to guide the diffusion-based molecular generation process - which is well-motivated and reasonable.
The ideas are presented in a logical and clear fashion, making the underlying motivations clear and their actual implementation in PromptDiff relatively easy to understand.
The evaluation results show that PromptDiff is capable of generating stable molecular structures and generate ligan molecules that can tightly bind to protein targets thanks to its "target-awareness".
Furthermore, a number of other properties - such as drug-likeness and diversity - of the generated molecules are enhanced compared to other diffusion-based methods with comparable synthetic accessibility.

**Weaknesses:**

Overall, the paper is written well and the presented evaluation results make the proposed PromptDiff scheme look promising.
There are nevertheless several concerns/questions that need to be addressed.
These are summarized in what follows.

1. It is reasonable to expect that the use of exemplars with high binding affinity as prompts may be more efficient than designing ligands from scratch and also may effectively steer the diffusion-based molecular generation process to make the generated molecules possess high binding affinity against the given protein target.
However, one concern regarding such an approach is its impact on the diversity of the generated molecules.
Interestingly, the authors show that PromptDiff is capable of enhancing/maintaining molecular diversity compared with other diffusion-based methods and many other non-diffusion schemes.
But it is not intuitive why that's the case, and it warrants further investigation and discussion about this aspect.

2. An interesting question - also somewhat relevant to the "diversity" of the molecules generated by using exemplar prompting - is how the generated molecules would perform if the original database lacks molecules with high binding affinity.
Even in such scenarios, would PromptDiff be capable of generating diverse molecules such that the generated molecules would include ligands with high binding affinity to the protein targets?
Would the performance of PromptDiff still outperform other techniques that do not use such exemplar prompting?

3. In the evaluation results, how were the "reference molecules" selected?
Please clarify, as this is not clearly described in the current manusscript.
Also at leat brief define (with proper citations) how QED, SA, Diversity metrics were evaluated, since there exist different schemes/tools for this purpose.

4. Since PromptDiff mainly aims to generate molecules with enhanced binding affinity to the target proteins, is the fact that it also enhances some other properties compared to other approaches simply a (somewhat unexpected) by product of the proposed diffusion-based generation scheme?
Or would one expect to similar outcomes for various other molecular optimization tasks/scenarios in general?
Please share relevant insights/intuition (if any).

5. In the ablation study, the paper discusses the impact of applying promopting without the pre-trained PMINet.
What conclusions can be drawn regarding the (relative) importance/efficacy of prompting using "good exemplars" vs. the use of the learned ligand-protein interactions captured in the interaction layer of PMINet (used in self-prompting)?

6. It appears that results in Table 4 are somewhat limited to draw a reliable conclusion (especially so for k).
There is no discussion regarding the impact of T.
Furthermore, the impact of these hyperparameters on the overall computational cost should be discussed.

7. Considering that multi-objective optimization is commonly required for molecular design, it is important to discuss how the proposed scheme would extend to the case when multiple molecular properties need to be optimized simualtaneously.

**Questions:**

Please see the questions and concerns summarized in the "Weaknesses" section.

---

### Official Review · Reviewer_DL9h · 2023-11-06

**Soundness:** 3 good
**Presentation:** 3 good
**Contribution:** 2 fair
**Rating:** 3
**Confidence:** 4

**Summary:**

The proposed method presents PromptDiff, a prompt-based approach to generate 3D molecules via diffusion models. This method uses pre-trained binding affinity signals and exemplar prompting and self-prompting techniques to generate ligands. Empirical results show improvement of generating drugs with high binding affinity.

**Strengths:**

1. The first one to adopt prompting techniques for 3D molecule drug designs.
2. Introduce two novel prompting techniques.
3. Empirically show some improvement in generating ligands with high-affinity scores.

**Weaknesses:**

1. The method of this work lacks novelty. Prompting is a common technique in NLP, and diffusion models are one of the most popular models in image generation. The paper would benefit more if the author could explain the insights behind choosing prompting and diffusion models as the main method.

2. The pipeline has several steps, retrieving and prompting. How each step affects the molecules is not clear. For example, a model is easily affected by the selection of retrieved molecules, but the model is robust to selecting top-1 affinity molecules. This part of the argument is not inconsistent and needs empirical evidence.

**Questions:**

Please see *weaknesses* section.

---

### Meta-Review · Area_Chair_tTZa · 2023-12-05

**Metareview:**

This paper investigates the problem of 3D molecular generation via a diffusion model with prompting technique. Despite the majority of the reviewers being negative about the acceptance of the paper, only one reviewer offered a positive review. However, the positive review may not be strong enough to support acceptance, as several concerns raised by the reviewer (e.g., diversity of generated molecules) remain unresolved (the authors did not prove rebuttals for the reviewers' concerns). Hence, AC cannot suggest acceptance at the current form.

**Justification For Why Not Higher Score:**

The authors did not prove any rebuttals for the reviewers' concerns.

**Justification For Why Not Lower Score:**

N/A

---

### Decision · Program_Chairs · 2024-01-16

Reject